# Rilotumumab Resistance Acquired by Intracrine Hepatocyte Growth Factor Signaling

**DOI:** 10.3390/cancers15020460

**Published:** 2023-01-11

**Authors:** Fabiola Cecchi, Karen Rex, Joanna Schmidt, Cathy D. Vocke, Young H. Lee, Sandra Burkett, Daniel Baker, Michael A. Damore, Angela Coxon, Teresa L. Burgess, Donald P. Bottaro

**Affiliations:** 1Urologic Oncology Branch, Center for Cancer Research, National Cancer Institute, National Institutes of Health, Bethesda, MD 20892, USA; 2Amgen, Inc., Thousand Oaks, CA 91320, USA; 3Molecular Cytogenetics Core Facility, Mouse Cancer Genetics Program, Center for Cancer Research, National Cancer Institute, National Institutes of Health, Frederick, MD 21702, USA

**Keywords:** acquired drug resistance, glioblastoma, hepatocyte growth factor, Met, rilotumumab

## Abstract

**Simple Summary:**

Drug resistance is a long-standing impediment to effective systemic cancer therapy and acquired drug resistance is a growing problem for new therapeutics that otherwise have shown significant successes in disease control. Hepatocyte growth factor (HGF)/Met receptor pathway signaling is frequently involved in cancer and is widely targeted in drug development. We found that resistance to the HGF-neutralizing antibody drug candidate rilotumumab in glioblastoma cells was acquired through HGF overproduction and misfolding, which led to stress-response signaling and redirected transport inside cells that sequestered rilotumumab and misfolded HGF from native HGF and activated Met receptor. Resistant cells were more malignant but retained their sensitivity to Met kinase inhibition and gained sensitivity to inhibition of stress signaling and cholesterol biosynthesis. Defining this rapidly acquired, multisystem scheme improves our understanding of drug resistance and suggests strategies for early detection and intervention.

**Abstract:**

Drug resistance is a long-standing impediment to effective systemic cancer therapy and acquired drug resistance is a growing problem for molecularly-targeted therapeutics that otherwise have shown unprecedented successes in disease control. The hepatocyte growth factor (HGF)/Met receptor pathway signaling is frequently involved in cancer and has been a subject of targeted drug development for nearly 30 years. To anticipate and study specific resistance mechanisms associated with targeting this pathway, we engineered resistance to the HGF-neutralizing antibody rilotumumab in glioblastoma cells harboring autocrine HGF/Met signaling, a frequent abnormality of this brain cancer in humans. We found that rilotumumab resistance was acquired through an unusual mechanism comprising dramatic HGF overproduction and misfolding, endoplasmic reticulum (ER) stress-response signaling and redirected vesicular trafficking that effectively sequestered rilotumumab and misfolded HGF from native HGF and activated Met. Amplification of *MET* and *HGF* genes, with evidence of rapidly acquired intron-less, reverse-transcribed copies in DNA, was also observed. These changes enabled persistent Met pathway activation and improved cell survival under stress conditions. Point mutations in the HGF pathway or other complementary or downstream growth regulatory cascades that are frequently associated with targeted drug resistance in other prevalent cancer types were not observed. Although resistant cells were significantly more malignant, they retained sensitivity to Met kinase inhibition and acquired sensitivity to inhibition of ER stress signaling and cholesterol biosynthesis. Defining this mechanism reveals details of a rapidly acquired yet highly-orchestrated multisystem route of resistance to a selective molecularly-targeted agent and suggests strategies for early detection and effective intervention.

## 1. Introduction

Acquired drug resistance is a long-standing problem of cancer therapeutics. For example, chemoresistance to the DNA-alkylating agent temozolomide occurs in >90% of recurrent glioblastoma multiforme (GBM), often through increased expression of *MGMT* which encodes an alkyltransferase capable of repairing the DNA damage. The issue has become even more vexing with the development of highly selective agents such as those targeting the epidermal and hepatocyte growth factor (EGF and HGF, respectively) pathways. Acquired resistance to gefitinib or erlotinib in lung adenocarcinomas is a prevalent response to prolonged treatment and occurs through HGF pathway activation and other molecular mechanisms ([1,2,3,4,5,6,7,8,9], reviewed in [10,11]). Anticipating acquired resistance and understanding its basis should aid in the development of clinical strategies to prevent or circumvent its occurrence.

HGF, through its receptor tyrosine kinase Met, regulates mitogenesis, motogenesis, and morphogenesis in a range of cellular targets during development and homeostasis [12]. HGF/Met signaling also contributes to oncogenesis and tumor progression in many human malignancies, including GBM. The *HGF* and *MET* genes are expressed in human glioma and medulloblastoma, where their increased relative abundance frequently correlates with tumor grade, tumor blood vessel density and poor prognosis [12]. The *HGF* and *MET* genes are collectively altered (amplified, overexpressed and/or mutated) in published and provisional GBM datasets compiled by The Cancer Genome Atlas (TCGA) Research Network in 7.5% of 206 cases [13] and 18% of 291 [14] and 528 cases [15]; 2% of the latter 819 cases show concomitant overexpression or amplification of both genes [14,15]. Overexpression of *HGF* and/or *MET* in brain-tumor-derived cells enhances their tumorigenicity and growth, and inhibition of HGF or Met in experimental tumor xenografts suppresses tumor growth and angiogenesis [16,17,18,19]. Elevated levels of HGF protein in human cerebrospinal fluid are associated with mortality and GBM recurrence [20]. *MET* expression in GBM-derived cultures enriched for stem and progenitor cells was associated with mesenchymal and proneural gene signature GBM subtypes and with invasive and stem-like phenotypes [21]. Consistent with the suspected role of HGF in GBM progression, potent and highly selective antagonists of HGF–Met and Met–ATP binding interactions significantly inhibited subcutaneous and intracranial brain tumor growth in mice [22,23,24]. Although both paracrine and autocrine HGF are believed to enhance GBM growth, autocrine HGF/Met signaling in GBM cell lines reliably predicted sensitivity to pathway inhibition in vivo [25].

Rilotumumab (AMG102) is a fully human neutralizing monoclonal antibody against HGF that potently inhibited the growth of GBM-derived cell lines such as U87 MG, which has HGF/Met autocrine signaling, in culture and in tumor xenografts as a single agent. In addition, rilotumumab enhanced the efficacy of temozolomide or docetaxel in U87 MG tumor-bearing mice and enhanced the killing effect of ionizing radiation on U87 MG cells in vitro and in vivo [22,25,26,27,28]. Rilotumumab was found to be safe and well tolerated in phase I human clinical trials [29] and has been tested in multiple phase II clinical trials [30]. In anticipation of acquired resistance to rilotumumab in GBM, models were generated by growing U87 MG cells in maximally effective drug concentrations and by escalating dose treatment of mice implanted with U87 MG cells. Rilotumumab-resistant cell lines and tumors remained sensitive to a selective small molecule Met tyrosine kinase inhibitor, and therefore were dependent on HGF/Met signaling, which occurred through HGF overproduction and misfolding-induced ER stress-response signaling and rilotumumab uptake-driven subversion of vesicular trafficking that secluded Met activation from rilotumumab.

## 2. Materials and Methods

### 2.1. Reagents and Cell Culture

Rilotumumab was prepared by Amgen Inc. (Thousand Oaks, CA, USA) at a stock concentration of 30 mg/mL. An IgG2 antibody (Amgen Inc.) was used as an isotype control. Recombinant human HGF/NK1 (HGF variant 5) was prepared as described [31]. All three agents were stored at −80 °C until dilution. AMG517 (Ref. [32], identified therein as compound #22) was formulated in soybean oil at a concentration of 10mL/kg. Simvastatin (HMG-CoA reductase) and the PERK inhibitor GSK2656157 were purchased from SelleckChem.

The U87 MG-derived, rilotumumab-resistant cell line (designated U87 MG/HNR for HGF Neutralization Resistant) was generated by growing U87 MG cells (also known as HTB-14, obtained from ATCC, Manassas, VA, USA) continuously in normal growth media [2] plus rilotumumab at 100 nM for a period of 5 days followed by 115 days of rilotumumab at 600 nM for a total treatment period of 120 days. Aliquots of these cells were frozen for subsequent subculture and analysis. Subcultures of U87 MG/HNR grown in 1 µM rilotumumab showed no additional phenotypic changes. Cells grown for experiments were maintained in rilotumumab unless otherwise noted. For the in vivo experiments, U87 MG parental and U87 MG/HNR cell lines were cultured in DMEM High Glucose media (Gibco) with 10% fetal bovine serum (FBS Gibco) and 1X L-glutamine (Gibco). Other cell lines used were also obtained from ATCC (Manassas, VA, USA).

### 2.2. Quantitative Immunoassays

HGF, Met, and phosphoMet content in cell lysates, blood plasma, or conditioned media were determined using 2-site electrochemiluminescent immunoassays as described previously [31]. Met activation as indicated by phosphoMet content in cell lysates included parallel detection with anti-receptor antibodies and specific anti-phospho-receptor antibodies or the monoclonal anti-phosphotyrosine (anti-pY) antibody clone 4G10. Cultured cells were serum-deprived for 16–24 h in the presence or absence of various agents as noted in the text prior to stimulation for 20 min with HGF (1 nM) at 37 °C alone or in combination with rilotumumab or compound A at the indicated concentrations. The cells were extracted with ice cold buffer containing non-ionic detergent, protease and phosphatase inhibitors; cleared extracts were applied to plates containing immobilized HGF or Met capture antibody and detected with anti-HGF, anti-Met or anti-pY. For some of the experiments described in Figure 7 and Figure 8, TX-100-insoluble cell extracts were solubilized in the same buffer containing 1% SDS and diluted 10-fold before measuring protein, HGF, rilotumumab, Met or pMet content.

All measurements were made on triplicate samples; the protein content of all samples was determined prior to immunoassay. All samples were adjusted to 0.5–0.7 mg/mL prior to immunoassay and all immunoassay values obtained were normalized to actual total protein values. HGF and Met content assays include purified recombinant reference standards for absolute quantitation. GraphPad Prism software version 5.0 was used for all statistical analyses. Protein content values were interpolated from standard curves by nonlinear regression analysis. All other statistical tests are described in the Results and Figure Legends.

### 2.3. Cell Proliferation and Anchorage Independent Growth Assays

For proliferation assays, U87 MG cell lines (5 × 10^4^ cells per well) were seeded in 6-well culture plates in quadruplicate. The cell number per well was measured after 2, 4, 5 and 6 days of growth by removing cells with trypsin, collecting via centrifugation and counting suspended cells in triplicate in an automated cell counter. Assays for colony formation in soft agar were performed as described [31]; colonies were treated with 3,4,5-dimethyl thiazole-2-yl-2,5-diphenyltetrazolium bromide and the dye product was eluted with MeOH and quantitated by absorbance at 590 nm. Differences between mean values were determined by an unpaired *t*-test with Welch’s correction using GraphPad Prism v5.0.

### 2.4. Tumorigenicity Assays

All experiments involving animals were performed in accordance with NIH Guidelines for Care and Use of Laboratory Animals, using institutionally reviewed and approved protocols at Amgen (Animal Protocol 2015-01243; Thousand Oaks, CA, USA) or the National Cancer Institute (NIH Animal Study Protocol UOB-009; Bethesda, MD, USA). U87 MG and derived cell lines were injected subcutaneously into athymic nude mice (Jackson Laboratories and Harlan Laboratories; n = 10 per group) and tumor volumes were measured at regular intervals as described previously [26,31]. Mice were treated with rilotumumab by intraperitoneal injection every two days or with AMG517 by oral gavage every day at doses indicated in the text. Animals were sacrificed and tumors were removed for histopathology by conventional methods. Tumor growth curves were fitted by regression analysis (R^2^ > 0.95) using GraphPad Prism software version 5.0. Measurement of plasma HGF protein levels was performed using the two-site electrochemiluminescent method [31], or by SDS-PAGE, electrophoretic transfer to PVDF membranes and immunoblotting using a polyclonal antibody directed against the human HGF amino-terminal sequence (Santa Cruz Biotechnology SC-1357). Levels of HGF in serum in U87 MG/HNR tumor-bearing mice were determined using either the Quantikine ELISA Human HGF Immunoassay (R&D Systems) or electrochemiluminescent immunoassay as described above. Data are expressed as mean ± SEM. A one-way ANOVA followed by Dunnett’s post hoc testing was used to determine statistically significant differences. Other statistical analyses, curve fitting and IC_50_ determinations were performed using GraphPad Prism v5.0 as indicated in the Results and Figure Legends.

### 2.5. CGH and mRNA Profiling Arrays and Comparisons to TCGA Datasets

Genomic DNA was purified (DNeasy Blood and Tissue Kit, Qiagen, Inc., Valencia, CA) and mixed with either Cy3- or Cy5-labeled primer solution (0.5 OD; Cy-labeled primer, Trilink BioTechnologies, San Diego, CA) in 125 mM Tris HCl (pH 6.8) and 12.5 mM MgCl_2_. After incubation at 99 °C for 10 min, reactions were brought to a total volume of 50 mL with 1 mM dNTPs and 50 U Exo-Klenow (Life Technologies, Carlsbad, CA). Reactions were incubated for 2 h at 37 °C prior to termination by heating to 65 °C for 10 min. Labeled product was purified using ethanol precipitation and quantified using a Nanodrop ND-1000 (Nanodrop Products, Wilmington, DE, USA).

Arrays were hybridized following the Oligonucleotide Array-Based comparative genomic hybridization (CGH) for Genomic DNA Analysis (Agilent Technologies, Santa Clara, CA, USA), with modifications. Briefly, 10 µg of Cy5-labeled experimental sample was mixed with 10 µg of Cy3-control DNA (pool of 40 normal male donor gDNAs) and incubated in a hybridization cocktail according to the manufacturer’s instructions. Following incubation, samples were hybridized against the SurePrint G3 Human CGH Microarray 2 × 400K for 40 h at 65 °C, rotating at 20 rpm. Samples were washed according to the manufacturer’s instructions and scanned at 2 µm using 100% PMT for Cy3 and 50% PMT for Cy5 on an Agilent DNA Microarray Scanner Model G2505C. Scanned data were feature extracted using Agilent Feature Extraction v10, and .txt files were imported to Agilent Genomic Workbench software to generate log_2_ ratio scores and aberration calls.

Total RNA was purified (RNeasy Mini Kit, Qiagen, Inc., Valencia, CA, USA) and profiled after the Agilent One-Color Microarray-Based Gene Expression Analysis Protocol, hybridizing to a customized Agilent Human Whole Genome V2 Microarray (Agilent Technologies, Santa Clara, CA, USA) with every reporter replicated at least four times (AMADID 026822). Array data were extracted using Agilent Feature Extraction Software (version 10.7) and imported into Rosetta Resolver software (version 7.2.2) for analysis.

Following background correction and normalization of microarray data, a difference set of significant mRNA expression changes between the parental and resistant cell lines was derived (defined as >1.17-fold and *p* < 1.00 × 10^−4^; 11,523 probe IDs) and a subset of 7688 genes that were significantly modulated 1.5-fold or greater (Appendix A) were uploaded for Ingenuity Pathway Analysis (IPA) core analysis with either User Data Set or the Agilent Whole Genome Microarray 4 × 44k v2 probe set as reference set; both direct and indirect relationships were included. Default IPA settings were used for Networks (Interaction), Data Sources (all), Species (all), Tissues and Cell Lines (all) and Mutations (all). Probability (*p*) values for significant overlap (*p* < 0.05) with gene sets defining Networks, Biofunctions, Canonical Pathways and Upstream Regulators were derived using the right-tailed Fisher’s Exact test (single correlations) or the Benjamin–Hochberg Multiple Testing Correction (grouped correlations). All gene names and symbols listed conform to current HUGO Gene Nomenclature Committee convention.

A comparison of the expression profiling dataset (Appendix A) with the TCGA GBM datasets [13,14,15] was performed using tools available through the cBioPortal [33,34].

### 2.6. Fluorescence In Situ Hybridization (FISH) Karyotyping

Multi-color FISH (M-FISH) karyotyping was performed in the Comparative Molecular Cytogenetic Core Facility at the National Cancer Institute. Briefly, chromosome preparations were obtained from established drug-resistant cell cultures, which were suspected to have generated double-minute chromosomes, by standard procedures. Slides were prepared and incubated overnight for use in FISH analysis of *MET* and *HGF* genes. For the detection of the *HGF* gene BAC Clone RP11-552M24 (Empire Genomics, Buffalo, NY, USA) was used with CEP 17 (Vysis, Abbott Molecular, Chicago, IL, USA) for verification of the location of the BAC clone. To analyze the *MET* gene, a DNA probe from Kreatech was used. Hybridization was carried out in a humidity chamber at 37 °C for 16 h according to standard protocols. The post-hybridization rapid-wash procedure was used with 0.4 × SSC at 72 °C for 4 min. Detection was carried out following the manufacturer’s protocol. Spectral images of the hybridized metaphases were acquired using an SD301 SpectraCubeTM system (Applied Spectral Imaging Inc., Carlsbad, CA, USA) mounted on an epi-fluorescence Axioplan 2 microscope (Zeiss, Oberkochen, Germany) using Spectral Imaging 6.0 acquisition software (Applied Spectral Imaging Inc., Carlsbad, CA, USA).

### 2.7. Real-Time Quantitative PCR

RNA was isolated from cell pellets from subconfluent cell cultures using the RNeasy kit (Qiagen, Valencia, CA) following the manufacturer’s instructions. Total RNA (20 ng) was reverse-transcribed with random primers to cDNA using the SuperScript III First-Strand Synthesis System (Invitrogen, Carlsbad, CA, USA). Variant-specific HGF primers were designed with the following DNA sequences: CV1: ACGAACACAGCTTTTTGCCTTC; CV1A: CCATGATACCACACGAACAC; CV2: CACACGAACACAGCTATCGG; CV3: CTGAACACTGAGGAATGTCAC; CV4A: CCCACATGGCATTCAGGTT; CV5: CCATGGTGCTATACTCTTGAC; CV6: GAAGTTCACCATCAGTTGAGAG; CV7: CTTGACCTTGGATGCATTCAG. These primer pairs distinguish the HGF variant cDNAs as follows: CV1A/CV3 (all variants), CV1/CV3 (variants 1, 2 and 5), CV2/CV3 (variants 3 and 4), CV1/CV4A (variant 5), CV5/CV7 (variants 1 and 3), CV5/CV6 (variants 2 and 4). Real-time quantitative PCR was performed with SYBR-Green PCR Master Mix (Applied Biosystems, Foster City, CA, USA) using an ABI 7000 real-time PCR system (Applied Biosystems) following the manufacturer’s protocols. All reactions were run in triplicate using *PPIA*, *GUSB* and *HPRT* as internal control genes. The relative level of gene expression was evaluated using the delta-delta CT method.

### 2.8. SDS-PAGE, Immunoblotting, HGF Affinity Chromatography and Ultrafiltration

SDS-PAGE was performed by conventional methods using precast gels (BioRad, Hercules, CA, USA). Electrophoretic transfer to PVDF membrane (Millipore, Burlington, MA, USA) and immunoblot analysis was performed using antibodies identified in the text and figure legends. Heparin-Sepharose CL-6B (GE Healthcare Life Sciences) affinity purification of HGF proteins was performed by batch loading conditioned media for 16 h at 4 °C, followed by stepwise elution with PBS containing increasing NaCl concentrations as listed in the text and figure legends. The 0.8 M NaCl fraction was further subjected to dialysis against PBS for 16 h at 4 °C, followed by centrifugal ultrafiltration using a 30 kDa Microcon concentrator (Millipore). The HGF content of both retentate and pass after filtration was determined by SDS-PAGE and immunoblotting using a standard curve made using purified recombinant HGF and NK1 proteins.

## 3. Results

### 3.1. HGF and Met Superabundance in Rilotumumab-Resistant U87 MG Cells

The HGF/Met dependent human GBM-derived cell line U87 MG was grown in continuous exposure to rilotumumab for 120 days to generate a cellular model of acquired resistance. For the first 5 days the rilotumumab concentration was 100 nM, which was then increased to 600 nM. Resistant cells (designated U87 MG/HNR for HGF Neutralization Resistant) displayed altered morphology in 2D culture relative to the parental cell line (Figure 1A,B). Resistant cells were modestly but consistently smaller, had fewer and shorter extended processes, and had fewer cell–cell interactions (visualized as cell clumping) than parental cells. Resistant cells were also noticeably less adherent to plastic or extracellular matrix substrata than the parental cell line. A relatively high number of floating cells also suggested an increased rate of cell death over the parental cell line. Indeed, U87 MG/HNR cells displayed substantially elevated activated caspase 3 relative to U87 MG, suggestive of increased apoptosis (Figure 1C; original immunoblot images for all figures are shown in Appendix A) despite a significantly higher rate of growth in culture (Figure 1D). Among the most distinguishing features of U87 MG/HNR was its >10,000-fold higher rate of HGF protein production compared to the parental U87 MG cells (Figure 1E). Met protein content and autophosphorylation level (phospho-Met) were also 8-fold and 80-fold higher than parental cell values, respectively (Figure 1F,G). The ratio of phospho-Met to total Met protein for U87 MG/HNR was similar to that of the normal mammary epithelial cell line 184B5 upon treatment with 1 nM exogenous HGF for 20 min at 37 °C (Figure 1G), indicating that Met maintained steady-state maximum kinase activity. Remarkably, U87 MG/HNR remained sensitive to the selective small-molecule Met kinase inhibitor AMG517 (Ref. [32], identified therein as compound #22); potent suppression of steady-state phospho-Met levels was observed in both U87 MG and U87 MG/HNR (Figure 1H), despite their significantly different total phospho-Met content (note the left vs. right *y*-axes scale difference).

### 3.2. U87 MG/HNR Tumorigenesis Is Rilotumumab-Resistant Yet Remains MET-Pathway Dependent

The tumor xenograft growth rate for the resistant cell line was significantly elevated over the parental line, even when one-sixth of the number of cells were implanted (Figure 2A). Plasma levels of human HGF correlated directly with U87 MG/HNR xenograft tumor mass, consistent with a prior report [22] for U87 MG (Figure 2B). In agreement with prior reports [22,26], tumor formation by U87 MG cells was potently suppressed by twice-weekly rilotumumab treatment; doses from 12 to 120 mg/kg caused complete regression of small, preformed xenograft tumors, as did a selective small-molecule Met kinase inhibitor AMG517 administered daily at 60 mg/kg (Figure 2C). As anticipated, rilotumumab treatment regimens that caused complete regression of U87 MG tumors had little impact on tumors arising from U87 MG/HNR cells, yet AMG517 was completely effective in blocking tumor growth by this cell line (Figure 2D). Similar to prior studies, rilotumumab sequestered human HGF in the blood of treated animals in a dose-dependent manner, indicating that target recognition by the antibody was not compromised (Figure 2E).

Consistent with a highly stressed cell state (further detailed below), rilotumumab-resistant U87 MG/HNR xenograft tumors exhibited chronic inflammation, regional necrosis and scattered blood pools (Figure 3A vs. Figure 3B). Higher (20×) magnification views reveal that higher tumor growth rates were directly correlated with mitotic indices in the resistant xenografts (Figure 3C). This magnification also revealed serous accumulations, as well as increased but poorly-organized extracellular matrix production, features not observed in matched-size U87 MG parental tumors (Figure 3D–F).

In parallel with generating a rilotumumab-resistant cell line in culture, a group of mice were implanted with U87 MG cells and xenograft tumors were allowed to grow for 3 weeks; mice were then treated for 30 days with low-dose rilotumumab (Figure 4A, bracket at top “4 mg/kg”). All mice showed some degree of tumor regression early in this first treatment period, but later some mice showed renewed tumor growth (Figure 4A). The rilotumumab dose was then increased (Figure 4A, bracket at top “40 mg/kg”) and two weeks later, persistent tumor growth in some mice suggested that they had acquired drug resistance (Figure 4A, brackets “R” vs. “S” at right). Eleven cell lines were established from these tumors, all of which showed significantly increased HGF production (Figure 4B; note *y*-axis “1” = 100-fold over the parental cell line) and increased phospho-Met content (Figure 4C) relative to the parental cell line. These cell lines also remained sensitive to Met inhibition by AMG517 with little deviation in IC_50_ concentration, suggesting the absence of *MET* mutation as a means of acquired resistance (Figure 4D).

### 3.3. HGF and MET Gene Amplification in U87 MG/HNR

Comparative genomic hybridization (CGH) array analysis revealed focal amplification of both *HGF* and *MET* genes in U87 MG/HNR cells, but not U87 MG cells (Figure 5A, yellow and green arrows, respectively). Closer views of individual probe intensity values and moving averages for U87 MG (Figure 5B,C, blue line) or U87 MG/HNR cells (Figure 5B,C, tan line) indicate the presence of multiple extra copies of each gene in the resistant cell line. cDNA sequences derived from *HGF* and *MET* mRNA transcripts were normal in both the parental and resistant cell lines (i.e., 100% identity with UniProt P08581-1 (*MET*) and UniProt P14210-1 (*HGF*).

Although the observed increases in gene copy number are consistent with the increased relative abundance of HGF and Met proteins in U87 MG/HNR, the extraordinary level of HGF protein production suggested that additional mechanisms of upregulation might also contribute. Disruption of an *HGF* gene promoter region termed DATE (for deoxyadenosine tract element) that acts as a transcriptional repressor and consists of 30 tandem deoxyadenosines has been reported to increase *HGF* expression in breast cancer [35]. DNA sequencing the DATE region of the *HGF* promoter in U87 MG and U87 MG/HNR revealed that it was of normal length in both cell lines (Figure 5D). The leiomyosarcoma cell line SK-LMS1, which also has autocrine HGF/Met signaling, and the clear cell renal cell carcinoma cell line UOK331 provided unambiguous positive and negative controls for DATE region truncation, respectively (Figure 5E).

Interestingly, not all probes in the CGH array within either the *HGF* or *MET* gene regions had (log_2_ ratio) intensity values indicative of gene amplification. In fact, probes that fell within introns had normal copy number values, probes within exons had uniformly elevated values and probes that fell on intron/exon borders had intermediate values such that a significant linear relationship was observed between relative CGH probe intensity and the proportion (%) of probe/exon overlap (Figure 6A,B, r^2^ = 0.8695 and 0.9920 for *HGF* and *MET*, respectively). This suggested the possibility that extra gene copies had been acquired by retrotransposition, a process with potentially diverse impacts on cancer development [36]. To our knowledge, this is the first evidence of *HGF* or *MET* somatic reverse-transcribed gene amplification linked to acquired drug resistance. If functional, acquired *HGF* and *MET* pseudogenes could increase the rate of encoded protein production by eliminating the need for RNA splicing, in addition to doing so by increasing template abundance. Moreover, apart from the functionality of their encoded proteins, the RNA transcripts of these pseudogenes could also act as decoys that undermine miRNA regulation [37,38]. The additional possibility that multiple copies of *HGF* and *MET* genes might amplify through a chronology of autonomous circular DNA replication, double-minute chromosome formation and chromosomal integration, as described for other cancer-associated gene amplification events [39,40,41], prompted us to perform fluorescence in situ hybridization (FISH) to visualize *HGF* and *MET* genes in the parental and resistant cell lines (Figure 6C,D). Consistent with CGH array results, U87 MG had normal *HGF* and *MET* chromosomal localization and copy number (Figure 6C), whereas U87 MG/HNR displayed homogenously staining regions suggestive of amplification within chromosome 7 and trisomy of chromosome 7 (Figure 6D). Uniquely present in U87 MG/HNR were numerous double-minute chromosomes that were positive for *HGF* or *MET* FISH probes (Figure 6E,F); these DNA fragments were visible within 3 weeks of rilotumumab treatment and increased with longer periods of drug treatment. To independently confirm the relationship between CGH probe intensities and the proportion of exon overlap, PCR primers were generated corresponding to the CGH probe regions of genomic *HGF* and *MET* DNA sequences. qRT-PCR was performed using these primers, and comparing relative PCR product abundance with the Agilent CGH probe intensity values produced a significant linear relationship (Figure 6G; *p* = 0.0109 for regression non-linearity). DNA sequencing of several of these PCR products further revealed complete continuity between adjacent exons for both genes (data not shown). Further evidence that a portion of HGF protein superproduction by U87 MG/HNR cells resulted from *HGF* gene amplification by reverse transcription was obtained by growing the resistant and parental cell lines in the presence of rilotumumab and in the presence or absence of the reverse transcriptase inhibitor azidothymidine (AZT) and measuring HGF protein production over time. As shown in Figure 6H, AZT treatment significantly reduced HGF production by U87 MG/HNR but not by the parental cells.

### 3.4. HGF Transcript Variant 5: A Possible but Unobserved Route to Resistance

Rilotumumab binds to a region encompassing the nascent amino terminus of the light (or beta) chain in the mature, 2-chain HGF protein, which undergoes conformational rearrangement and exposure upon proteolytic activation of the single-chain pro-HGF precursor [42,43,44]. Because the primary Met-binding epitope in HGF resides in the amino (N)-terminal and kringle 1 (K1) domains and because the NK1 protein (also known as HGF variant 5) is a naturally occurring and biologically active truncated HGF isoform [12 and references therein], NK1 production offers a theoretical route to rilotumumab resistance while sustaining Met signaling. This was confirmed by analyzing the rilotumumab dose–response of phospho-Met levels in 184B5 cells stimulated with purified recombinant full-length HGF or NK1 proteins. As shown in Figure 7A, NK1-induced Met activation remained high in the presence of rilotumumab at concentrations that completely suppressed activation by full-length HGF.

**Figure 7 cancers-15-00460-f007:**
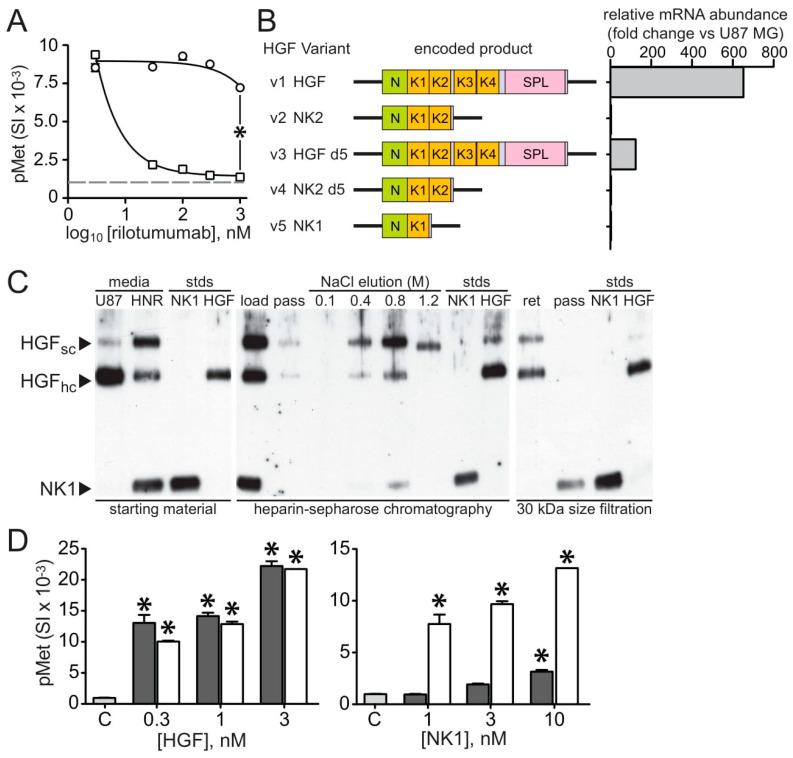
**Potential route of rilotumumab resistance by production of a non-neutralizable truncated HGF protein.** (**A**) Rilotumumab potently neutralizes Met autophosphorylation (pMet; mean signal intensity, SI ± SD, n = 3) in intact 184B5 cells induced by purified recombinant full-length HGF (squares) but not by the purified recombinant truncated HGF variant NK1 (circles). (**B**) Relative abundance (fold change vs. U87 MG; right) of all *HGF* mRNA transcript variants (listed and represented schematically at left) in U87 MG HNR as determined by quantitative PCR. (**C**) Reduced SDS-PAGE and immunoblot analysis of partial purification and size separation of HGF protein species present in U87 MG/HNR-conditioned media. Starting material (left panel) shows media (“media”) from U87 MG (“U87”) and U87 MG/HNR (“HNR”) contain single-chain (“HGF_sc_”) and mature HGF heavy-chain (“HGF_hc_”); an NK1-like protein (“NK1”) similar in size to purified recombinant NK1 (21 kDa, “stds”) is present only in the latter. HGF proteins from U87 MG/HNR media were partially purified using heparin-sepharose (center panel). Starting material before (“load”) and after (“pass”) application to the column are at left; the fraction (“NaCl elution”) eluting at 0.8 M NaCl was then dialyzed and sizing filtration (right panel) separated full-length (retained, “ret”) and truncated HGF (“pass”) forms. (**D**) Met activation (pMet) by partially purified full-length HGF (left panel, dark gray bars) and truncated HGF (NK1; right panel, dark gray bars) at indicated concentrations (nM) were compared with control media (“C”, light gray bars) and to purified recombinant HGF and NK1 protein standards (white bars). Asterisks indicate a significant difference from control (*p* < 0.05).

Steady-state levels of the five *HGF* transcript variants present in the parental and rilotumumab-resistant cell lines were compared by quantitative PCR. The left side of Figure 7B schematically depicts the domain structures of the *HGF* transcript variants (v): v1 and v3 encode full-length HGF protein forms that differ by the absence in v3 of five amino acid residues in the kringle 1 domain (v3 is thus known as delta 5 or d5); v2 is identical to v1 but truncated after the region encoding kringle 2 (also known as NK2); v4 is identical to v3 but truncated after kringle 2 (NK2 d5); and v5 encodes NK1, identical to the v1 product, but truncated after kringle 1 [12 and references therein]. Quantitative PCR analysis showed that the mRNA transcript levels for *HGF* v1 and v3, the full-length variants, were more than 600- and 100-fold more abundant in the resistant cell line relative to the parental, respectively (Figure 7B, right); v5 (encoding NK1) abundance was unchanged, suggesting that it was unlikely to have provided a route to resistance. Immunoblot analysis of 24-h-conditioned media from U87 MG/HNR confirmed that full-length HGF production was significantly higher than the parental cell line (Figure 7C, left side, “starting material”; note that U87 MG/HNR media was diluted >1000-fold relative to U87 MG media), but also revealed the presence of an HGF protein with similar molecular mass to a purified recombinant NK1 protein standard (“stds”) and that was recognized by an HGF antibody directed against the amino terminus, indicating that it contained the primary Met binding epitope. The precise amino-acid sequence and how this NK1-like protein was produced by U87 MG/HNR were unclear, but an assessment of its biological activity was undertaken.

Media conditioned by the resistant cell line was partially purified by heparin Sepharose chromatography. SDS-PAGE and immunoblot analysis (Figure 7C, center) showed that the majority of both full-length and truncated HGF proteins eluted with 0.8 M NaCl, consistent the presence of the amino-terminal domain containing the primary heparin binding site in the truncated form [12 and references therein]. Protein eluted in this fraction was dialyzed and subjected to ultrafiltration using a membrane with a 30 kDa cutoff; full-length HGF was retained, whereas the truncated, NK1-like HGF was not (Figure 7C, right side). The bioactivities of these HGF proteins were compared to each other and to purified, recombinant HGF and NK1 proteins by their abilities to stimulate Met activation in 184B5 mammary epithelial cells. As shown in the left panel of Figure 7D, Met activation (pMet) by partially-purified full-length HGF was indistinguishable from fully-purified recombinant HGF protein. In contrast, the truncated 21 kDa HGF form purified from U87 MG/HNR-conditioned media (Figure 7D, right side) had very little activity when compared with purified recombinant NK1 protein. This NK1-like protein was therefore unlikely to provide rilotumumab resistance.

### 3.5. Mechanistic Insights from Gene Expression Profiling and Pathway Analysis

A subset of 7688 genes that were significantly modulated 1.5-fold or greater up or down in the rilotumumab-resistant cell line relative to the parental cells (listed in Appendix A) were analyzed using Ingenuity Pathway Core Analysis software (IPA). Not surprisingly, the most significant IPA “Disease and Disorder” identified (IPA Summary, Appendix A, pp. 2–3) was “Cancer” (*p* value range: 8.35 × 10^−4^–1.20 × 10^−13^; number of modulated molecules in overlap: 4515) and the most significant IPA “Molecular and Cellular Function” identified was “Cellular Growth and Proliferation” (*p* value range: 7.12 × 10^−4^–2.79 × 10^−25^; number of modulated molecules in overlap: 2146). Twenty-six IPA “Canonical Pathways” were identified as having significant overlap with the dataset by right-tailed Fisher’s Exact test and the Benjamini–Hochberg (B–H) multiple test correction (Appendix A). These pathways could be logically grouped into 6 functional categories to provide a broader phenotypic portrait of acquired changes in U87 MG/HNR: (1) stress signaling; (2) immune-related; (3) cell proliferation and survival; (4) established cancer networks; (5) second messengers and nuclear receptors; and (6) nervous-system-related (Appendix A). The relative abundance of stress and immune-related inflammatory signaling pathways, and those related to proliferation, survival and established cancer networks (34% and 39% of pathways, respectively) is consistent with an aggressive GBM phenotype [45].

Signaling pathways identified using IPA suggested potential intracellular cellular processes and signaling pathways underlying acquired rilotumumab resistance. Significant overlap with pathways in the phenotypic portrait groups 3, 4 and 5 as defined above might be expected for many particularly aggressive tumor types (including GBM), and group 6 pathways are consistent with a cellular origin of GBM. Phenotypic groups 1 and 2 however—stress signaling and immune-related pathways—suggested processes and pathways by which autocrine HGF/Met activation could evade contact with rilotumumab.

### 3.6. A Coordinated, Multiplex Route to Rilotumumab Resistance

Transcriptional reprogramming downstream of endoplasmic reticulum (ER) stress signaling provides tumor cells with the needed proteostasis machinery and contributes to adaptation and cell survival even in the face of increased cell death [46]. Biochemical evidence of increased ER stress signaling included substantially increased levels of active caspase 3 (Figure 1C), PERK and calreticulin proteins (Figure 8A) in U87 MG/HNR relative to the parental cells. Increased total eIF2a protein and phospho-(ser51)-eIF2a were also observed in the resistant cell line (Figure 8B, left side), and phosphorylation of eIF2a at ser51 was inhibited by treatment with the selective PERK inhibitor GSK2656157 [47] (Figure 8B, right side). One potential connection between HGF superproduction and increased ER stress signaling was very predictable: full-length HGF is structurally complex, being comprised of 6 subdomains with 20 intrachain disulfide bonds. HGF cannot be expressed recombinantly in bacterial systems without denaturation and refolding, and even yeast systems (e.g., *P. pastoris*) provide low yields of active protein unless they are specifically engineered to overexpress protein disulfide isomerase(s) to facilitate the correct disulfide bond pairing required for proper folding [48]. To further investigate HGF misfolding, parental and resistant cells were extracted in buffer containing 1% Triton X-100 (TX100) non-ionic detergent, and insoluble cell material pelleted after centrifugation was extracted with the same buffer containing 1% SDS detergent. All samples were then dissolved in Laemmli (SDS) sample buffer for resolving by SDS-PAGE in the absence of a disulfide-bond-reducing agent such as dithiothreitol or beta-mercapto-ethanol (i.e., “non-reducing SDS-PAGE”). As shown in Figure 8C, subsequent immunoblotting for HGF revealed monomeric HGF (~90 kDa) in both cell lines, as well as an abundance of misfolded, disulfide-bonded HGF dimers (~180 kDa), trimers (~270 kDa) and lower levels of higher-order multimers in U87 MG/HNR (left-most lane) but not U87 MG (right lanes). The HGF multimers in U87 MG/HNR were present only in the cell material pelleted after TX100 extraction (Figure 8C, “pel”), an ER-rich fraction. In fact, low disulfide isomerase activity (and consequent HGF misfolding) is very likely to have been a positive selection factor for acquired resistance to rilotumumab; 9 out of 10 genes encoding thioredoxin-domain-containing ER-resident protein disulfide isomerases (*PDIA3, PDIA5, PDIA6, TMX1, TMX3, TMX4, TXNDC5, P4HB* and *ERP44)* showed reduced expression in U87 MG/HNR (−1.5 to −3.3-fold; Appendix A) relative to the parental line. Consistent with an ER-stress-induced unfolded protein response, the superproduction of misfolded HGF protein was associated with its retention inside cells; the overall ratio of intracellular to secreted HGF in U87 MG/HNR was completely reversed relative to the parental cells (Figure 8D).

**Figure 8 cancers-15-00460-f008:**
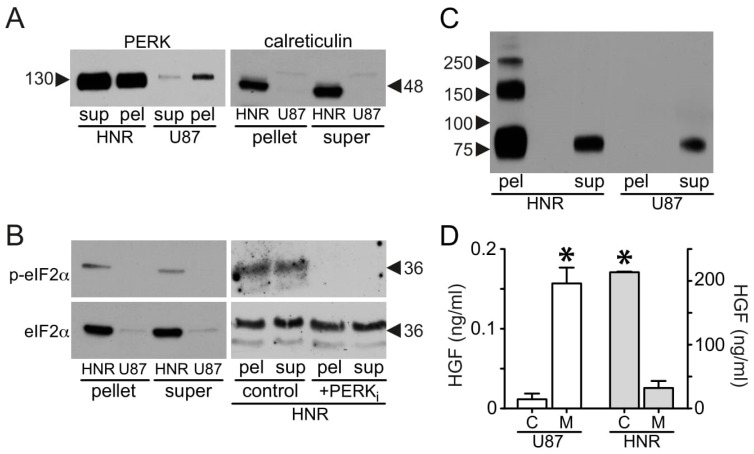
**Acquisition of ER stress through HGF protein misfolding in U87 MG/HNR.** (**A**) Reduced SDS-PAGE and immunoblot analysis of U87 MG (“U87”) and U87 MG/HNR (“HNR”) cell extracts for PERK (left panel) and calreticulin (right panel). Cells were extracted with ice-cold buffer containing TX-100 detergent, protease and phosphatase inhibitors before centrifugation to separate soluble (“sup”, left, or “super” right) and insoluble (“pel”, left, or “pellet”, right; these abbreviations also apply to panels (**B**,**C**)) cell fractions; insoluble fractions were subsequently solubilized in buffer containing 1% SDS prior to analysis. (**B**) Left, reduced SDS-PAGE and immunoblot analysis of U87 MG (“U87”) and U87 MG/HNR (“HNR”) TX-100 cell extracts for phospho-EIF2a (“p-eIF2a”; upper panels) and total EIF2α (lower panels); as in panel (**A**), TX-100 extracts were separated into soluble and insoluble fractions prior to analysis. Right, reduced SDS-PAGE and immunoblot analysis for phospho-EIF2a (upper panel) and total EIF2α (lower panel) for U87 MG/HNR (“HNR”) in the absence (“control”) or presence (“+PERK_i_”) of the selective PERK antagonist GSK2656157. As in panel (**A**), TX-100 extracts were separated into soluble and insoluble fractions prior to analysis. (**C**) Non-reduced (i.e., not treated with dithiothreitol or beta-mercaptoethanol so that disulfide bonds remain intact) SDS-PAGE and immunoblot analysis of U87 MG (“U87”) and U87 MG/HNR (“HNR”) cell TX100 extracts for HGF; as in panel (**A**), extracts were separated into soluble and insoluble fractions prior to analysis. (**D**) HGF protein concentration (ng/mL) in U87 MG cell (“U87”, white bars, left *y*-axis) lysates (“C”) and 24-h-conditioned media (“M”) and U87 MG/HNR cell lysate and media (“HNR”, gray bars, “C” and “M”, respectively, right *y*-axis). Mean values (ng/mL ± SD) from triplicate samples were normalized to total cell protein. Asterisks indicate a significant difference from control (*p* < 0.05). All the whole western blot figures can be found in the Appendix A.

Most significant among the immune-related IPA canonical pathway changes acquired with rilotumumab resistance was “Antigen Presentation” (75% pathway overlap, B–H multiple test *p* = 1.74 × 10^−3^, Appendix A). We suspected that substantial alterations in vesicular trafficking might effectively partition rilotumumab away from HGF and Met. Because lipid rafts are critical for many vesicular traffic systems [49,50] and these cholesterol-rich lipid regions are usually insoluble in TX100 detergent, we performed a series of experiments analyzing TX100 detergent soluble and insoluble cell lysate fractions of the resistant and parental cell lines for HGF, Met, pMet and rilotumumab content. Consistent with a role in acquired resistance, the rate of rilotumumab uptake by U87 MG/HNR was over 28-fold higher than that of the parental cells under identical conditions (Figure 9A). Moreover, whereas nearly all (88%) of the antibody taken up by the parental cell line was found in the TX100-soluble cell fraction (Figure 9A, left, gray bar), most rilotumumab uptake (~60%) in resistant cells was found in the TX100-soluble cell fraction (Figure 9A, right, white bar), consistent with a shift in rilotumumab uptake to a lipid-raft-dependent system. The cholesterol dependence of rilotumumab uptake was tested using the HMG CoA reductase inhibitor simvastatin. Although dose-dependent rilotumumab uptake by the parental cell line was unaltered by simvastatin treatment (Figure 9B, orange/yellow vs. gray/white bars), uptake by the resistant cell line was inhibited >98% (Figure 9C, orange/yellow vs. gray/white bars). Together these findings show that substantial quantitative and qualitative changes in rilotumumab uptake accompanied resistance.

Remarkably, increased rilotumumab uptake by the resistant cell line was also linked to Met activation. Met protein levels in the TX100-soluble and -insoluble cell fractions of U87 MG/HNR were similar in the absence of rilotumumab (Figure 9D, 0 nM rilotumumab). Rilotumumab addition did not change Met content in the TX100-soluble cell fraction (Figure 9D, gray bars), but was associated with a significant dose-dependent reductions of Met (Figure 9D, white bars) and phosphoMet (Figure 9E, white bars) in the TX100-insoluble cell fraction. Rilotumumab-driven Met and pMet loss from the TX100-insoluble cell fraction was blocked by simvastatin (Figure 9D,E, white vs. yellow bars), suggesting that rilotumumab uptake triggered a proteolytic process that was dependent on lipid-raft-mediated transport, potentially related to antigen presentation [51,52]. Although Met protein abundance was similar in the TX100-soluble and -insoluble fractions of untreated cells (Figure 9D, 0 nM rilotumumab), Met activation (pMet content) was three-fold higher in the TX100-soluble fraction (Figure 9E, 0 nM rilotumumab). Note that HGF protein in this fraction was not multimeric (Figure 8C) and thus was likely to be properly folded and capable of activating Met. Remarkably, rilotumumab treatment was associated with a significant and dose-dependent increase in Met activation in the TX100-soluble cell fraction and overall (Figure 9E, gray bars). This rilotumumab-driven increase in Met kinase activation was also blocked completely by simvastatin treatment (Figure 9E, orange vs. gray bars). These findings link U87 MG/HNR’s adaptive changes in the route of rilotumumab internalization to maintaining—and enhancing—HGF/Met pathway activation.

These new features underlying rilotumumab resistance—ER stress signaling and altered vesicular transport—also rendered U87 MG/HNR cells susceptible to inhibition of anchorage-independent growth by the PERK antagonist GSK2656157 and simvastatin, relative to the parental cell line (Figure 9F). Neither simvastatin nor GSK2656157 significantly inhibited soft agar colony formation by U87 MG, but these agents suppressed colony formation by U87 MG/HNR by 66% and 70%, respectively (*p* = 0.0024 and 0.0010), indicating an acquired dependence on these pathways and processes for rilotumumab resistance and enhanced HGF/Met-driven oncogenicity (Figure 9F).

## 4. Discussion

Acquired drug resistance to targeted single-agent therapies has become a serious obstacle to their long-term efficacy and use. For several malignancies such as non-small-cell lung cancer (NSCLC), breast, ovarian, and gastric cancers and melanoma, acquired resistance frequently involves activation of the HGF/Met pathway [1,2,3,4,5,6,7,8,9,53,54,55,56,57]. Aberrant HGF/Met pathway activation can result in increased tumor invasiveness, angiogenesis and metastasis, and is correlated with poor prognosis in many tumor types [58]. It is critical to understand the mechanisms by which acquired resistance develops, and preclinical studies have begun to focus on methods for detecting acquired resistance involving HGF/Met pathway activation as well as strategies to circumvent or remedy its manifestations [8,59,60,61,62,63,64,65].

Standard-of-care treatment for GBM is surgical resection followed by concurrent radiotherapy and chemotherapy with the DNA-alkylating agent temozolomide; this combination represents an improvement over prior approaches, but ultimately fails in almost all patients. Acquired resistance to radio- and chemotherapy for GBM is frequently related to the loss of DNA damage checkpoint mechanisms and/or enhanced expression of DNA repair machinery [66]. Together with several independent studies over the last decade, the genomic characterization of GBM undertaken by The Cancer Genome Atlas Research Network has helped to identify clinically-relevant subgroups of GBM that may benefit from therapeutics targeting growth factor pathways such as those of platelet-derived growth factor, EGF, HGF, and critical downstream effectors such as *PIK3CA* [13,14,15,45].

In anticipation that targeting HGF with rilotumumab in GBM may lead to acquired resistance, we developed the U87 MG-derived models described here. In U87 MG cells grown in the continuous presence of a maximally effective concentration of rilotumumab, amplification of both *HGF* and *MET* genes was associated with significantly increased HGF and Met abundance. CGH array data suggested that many of the extra gene copies lack introns and thus may have been acquired by retrotransposition. The roles of Alu and LINE-1 retrotransposons in cancer is an area of growing interest [36]; evidence of involvement in gene-amplification-associated acquired drug resistance and the remarkably short time span from deployment to impact expands current theory and warrants further investigation. Dramatically increased HGF secretion was also observed in 11 cell lines derived from rilotumumab-resistant U87 MG-derived tumor xenografts in mice, suggesting that HGF superproduction is an important route to resistance and may serve as a biomarker for its emergence in patients treated with HGF inhibitors. Although at first glance this appears to be a relatively simple mechanism relative to the spectrum and complexity of those reported for trastuzumab, cetuximab and bevacizumab, closer scrutiny indicated that the route to rilotumumab resistance was also multifaceted. Although overproduction of HGF variant 5 could have led to rilotumumab resistance, quantitative PCR analysis ruled against it and a truncated HGF protein in U87 MG/HNR-conditioned media showed little ability to activate Met.

Expression profiling, pathway analysis and confirmatory experiments pointed to a distinct and more complex mechanism. Of all 26 significant Ingenuity “Canonical Pathway” changes acquired by the resistant cell line, the “Antigen Presentation Pathway” had greatest overlap ratio (75%), followed closely by the “Endoplasmic Reticulum Stress Pathway” (71%). Confirmatory experiments showed that HGF superproduction specifically increased intracellular full-length HGF protein content, where a large portion of this full-length HGF protein was misfolded, multimeric and partitioned away from the monomeric soluble intracellular HGF protein. The misfolding of full-length HGF is very likely to have initiated ER stress signaling, enabling downstream events that promote cell survival despite increased apoptosis. These changes were also linked to significant quantitative and qualitative changes in rilotumumab internalization comprising a major shift to a lipid-raft-dependent transport system. This shift partitioned rilotumumab, misfolded HGF and a fraction of Met content destined for degradation to the same cell fraction, while sequestering soluble HGF and a distinct fraction of Met to a separate compartment that facilitated Met kinase activation.

The cellular origins of glioblastoma have been traced to neural stem cells, neural-stem-cell-derived astrocytes and oligodendrocyte precursor cells [67]. In the brain, all of these cell types have roles in antigen presentation; this characteristic may predispose GBM cells to adapt its underlying processes for drug resistance to mAbs. The autocrine HGF-Met dependence of U87 MG is consistent with Met as a functional marker of a glioblastoma stem cell phenotype [21] and the well-documented frequency of functional HGF-Met signaling in GBM [14,15,16,17,18,19,20,21,22,23,24,25]. ER stress signaling and lipid-raft-mediated rilotumumab uptake were essential components for the enhanced aggressive phenotype of the resistant cells and also presented new points of vulnerability; both simvastatin and PERK inhibitors became effective antagonists of anchorage-independent cell growth.

Remarkably, rilotumumab resistance in our cultured cell model was not associated with mutations in either *HGF* or *MET*, nor did it depend on alterations in other signaling pathways or mediators, since a selective small molecule Met TK inhibitor remained effective in blocking cell growth in vitro and tumorigenesis in vivo. The absence of target mutation and retention of target pathway dependence in our model of acquired rilotumumab resistance is thus unlike many other mechanisms of acquired drug resistance, where mutation of the gene encoding the drug target (e.g., *EGFR*), loss of prominent negative regulators downstream of the drug target (e.g., *PTEN*), or activation of alternative mitogenic pathways parallel to the target (e.g., *MET* in response to EGFR inhibitors) are prevalent means of restoring tumor cell proliferative and invasive activities. Our model is generally similar to others in that restoration of signaling via the PI3K and MAPK pathways was achieved, whether resistance was acquired to neutralizing antibodies [reviewed in [68]] or small TK inhibitors [54,55,56,57,69,70,71]. Despite the apparent diversity of the subcellular systems involved in the mechanism of acquired resistance to rilotumumab, its rapid development suggests that GBM cells in which HGF is an important oncogenic driver are predisposed to its adaptation as a route to survival. Measuring HGF/Met pathway activity in GBM patients is a logical basis for selecting those most likely to benefit from HGF/Met-targeted therapeutics; the results presented here further suggest that monitoring Met pathway activity and HGF production in those patients could provide early indications of acquired resistance, that Met kinase inhibitors may still be efficacious when resistance occurs and that targeting the other critical mediators of resistance identified here may provide effective alternative or combinatorial treatments.

## 5. Conclusions

Acquired drug resistance obstructs the effective treatment of many cancers and occurs through various mechanisms, often involving the elimination of drug from target cells or new defects in proto-oncogenes or tumor suppressors that revitalize essential growth and survival signaling pathways. Here we report a novel route to acquired resistance. In glioblastoma cells that require autocrine hepatocyte growth factor (HGF)/Met signaling for proliferation and survival, resistance to the HGF neutralizing monoclonal antibody rilotumumab was acquired through a complex interplay of several intracellular systems: (1) *HGF* and *MET* gene amplification and HGF protein super-production; (2) downregulation of ER-resident disulfide isomerases contributing to significant HGF protein misfolding; (3) induction of ER stress-response signaling and intracellular HGF and Met protein retention; and (4) dramatically increased rilotumumab uptake and degradation through a shift to caveolar endocytosis and activation of antigen presentation pathways. Together, these changes provided intracellular seclusion of properly folded HGF and Met proteins from rilotumumab and maintained HGF/Met signaling dependence for cell growth and survival.

Unlike other acquired drug-resistance mechanisms, mutation of the gene encoding the drug target, loss of critical negative regulators downstream of the drug target, and/or activation of alternative mitogenic pathways parallel to the target were not observed. Despite the number and diversity of the subcellular systems involved, resistance developed rapidly in GBM cells in which HGF is an important oncogenic driver. Defining this mechanism also revealed targetable co-acquired dependencies for survival in resistance cells: cholesterol synthesis needed for caveolar uptake and ER-stress signaling as well as continued sensitivity to small-molecule Met tyrosine kinase inhibitors. These findings suggest strategies for the early detection of this form of resistance and for effective intervention.

## Figures and Tables

**Figure 1 cancers-15-00460-f001:**
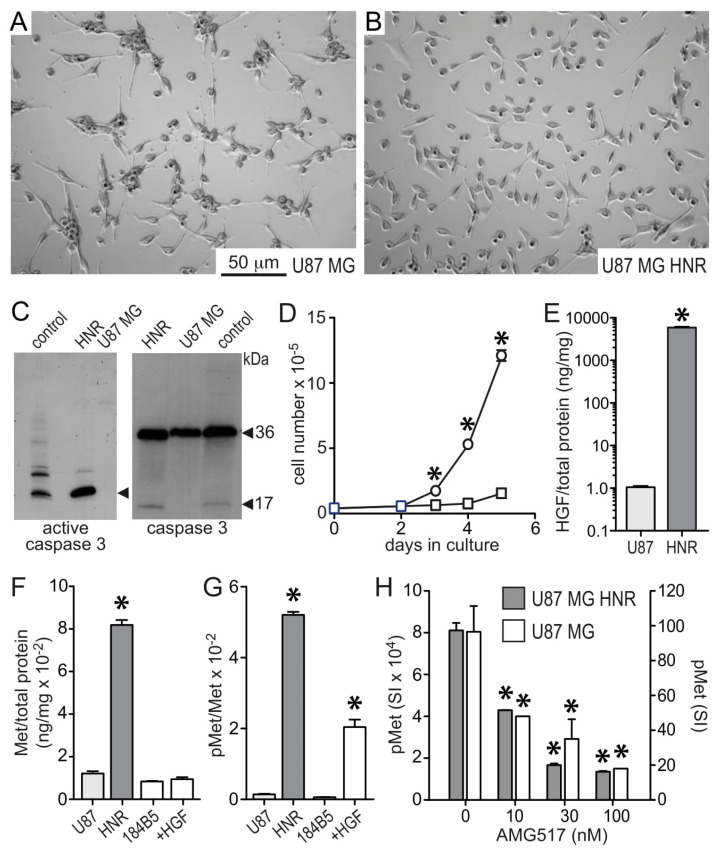
**U87 MG and U87 MG/HNR cell morphology, proliferation, HGF, Met and phospho-Met content.** Light micrographs of U87 MG (**A**) and U87 MG/HNR cells (**B**) in log growth phase. (**C**) Reduced SDS-PAGE and immunoblot analysis of U87 MG and U87 MG/HNR (“HNR”) cell extracts for activated caspase 3 (17 kDa, left panel) and total caspase 3 protein (36 kDa, right panel). Lanes marked “control” contain positive control proteins for active or intact caspase 3 provided by the antibody manufacturer. (**D**) Growth rates of U87 MG (squares) and U87 MG/HNR cells (circles) in culture (mean cell number ± SD, n = 3). (**E**) Secreted HGF protein (mean ng/mg total protein ± SD, n = 3) present in 24 h media from cultured U87 MG (light gray bar, left) or U87 MG/HNR cells (dark gray bar, right). (**F**) Total Met protein content (mean ng Met/mg total protein ± SD, n = 3) of U87 MG (light gray bar), U87 MG/HNR (dark gray bar) and serum-deprived 184B5 normal human mammary epithelial cells (white bars) in the absence (left) or presence of 1 nM HGF (right). (**G**) Phospho-Met (pMet) content (mean phosphoMet/total Met signal intensity ratio per mg total protein ± SD, n = 3) of U87 MG (light gray bar), U87 MG/HNR (dark gray bar) and serum-deprived 184B5 cells (white bars) in the absence (left) or presence of 1 nM HGF (right). (**H**) Phospho-Met (pMet) content (mean phosphoMet/total Met signal intensity ratio per mg total protein ± SD, n = 3) of U87 MG/HNR (gray bars, left *Y*-axis) and U87 MG (white bars, right *Y*-axis) in the absence (0) or presence of indicated concentrations (nM) of AMG517, a selective small-molecule Met kinase inhibitor. Asterisks indicate a significant difference from control (*p* < 0.05). All the whole western blot figures can be found in the Appendix A.

**Figure 2 cancers-15-00460-f002:**
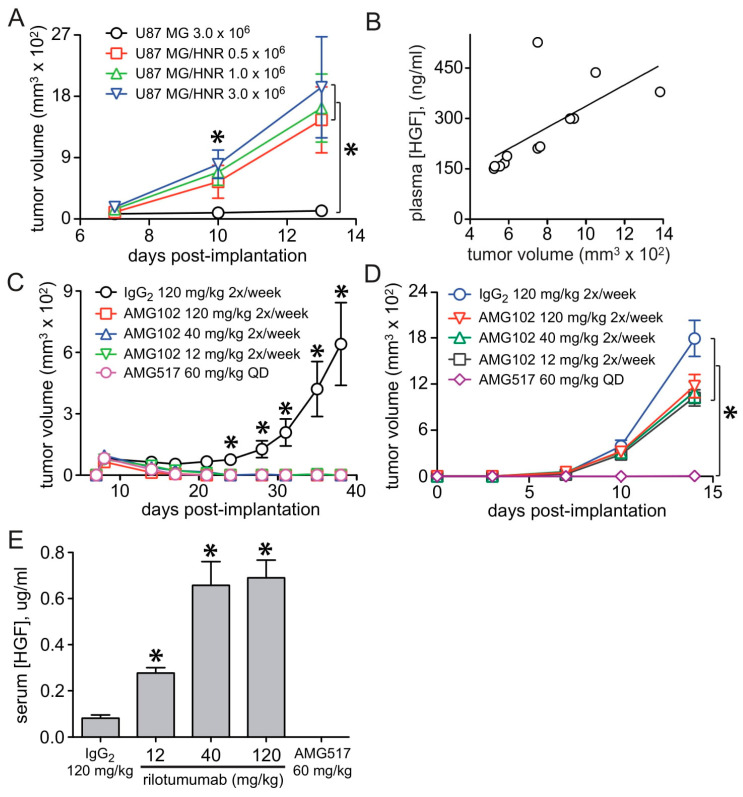
**Rilotumumab-resistant U87 MG/HNR cell xenograft growth remains HGF-pathway dependent.** (**A**) Tumor xenograft growth (mean tumor volume ± SEM, n = 10) in mice implanted with U87 MG (circles, 3 × 10^6^ cells/animal) or U87 MG/HNR cells (squares, triangles and inverted triangles) at 0.5, 1 or 3 × 10^6^ cells/animal, respectively. (**B**) Mean plasma HGF concentration (ng/mL) vs. tumor volume for mice implanted with U87 MG/HNR cells. (**A**,**C**) U87 MG tumor xenograft (5 × 10^6^ cells/mouse) growth in mice (mean tumor volume ± SEM, n = 10) treated with control IgG (black circles) or rilotumumab (AMG102) at 12 (green inverted triangles), 40 (blue triangles) or 120 mg/kg (red squares) or compound A at 60 mg/kg (violet circles). (**D**) U87 MG HNR xenograft growth (0.5 × 10^6^ cells/mouse, mean tumor volume ± SEM, n = 10) in mice treated with control IgG (circles) or rilotumumab (AMG102) at 12 (squares), 40 (triangles) or 120 mg/kg (inverted triangles) or the small-molecule kinase inhibitor AMG517 at 60 mg/kg (diamonds). (**E**) Mean (±SD) serum HGF concentration (µg/mL) at study termination in U87 MG/HNR tumor-bearing mice treated as in panel (**D**). Asterisks indicate a significant difference from control (*p* < 0.05).

**Figure 3 cancers-15-00460-f003:**
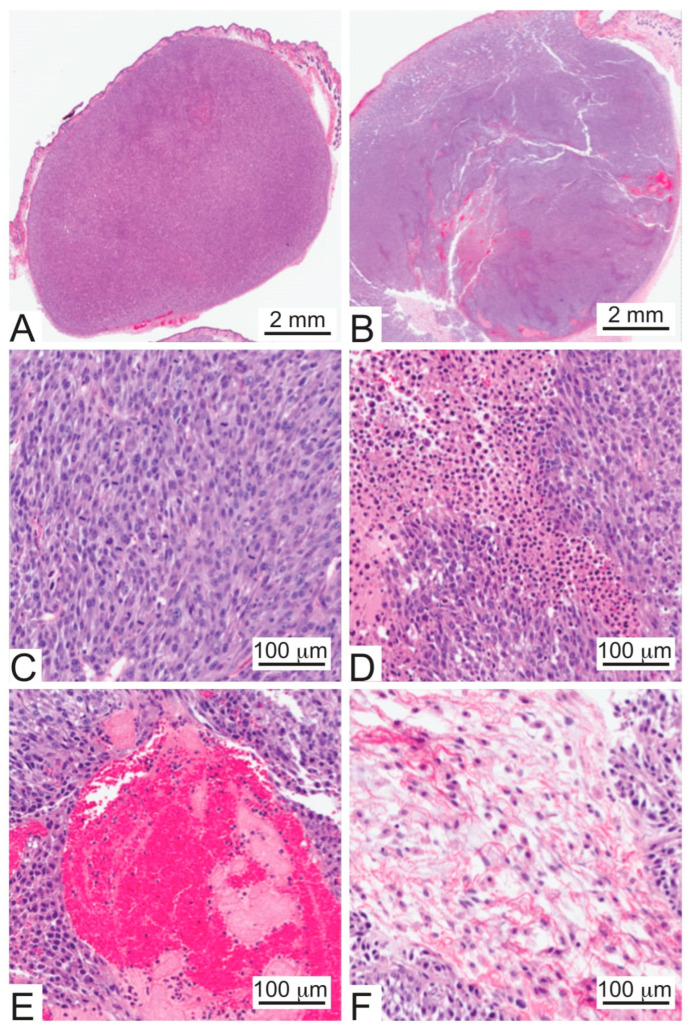
**Representative histology of U87 MG and U87 MG/HNR xenograft tumors.** Low magnification (1×) of representative tumors derived from (**A**) U87 MG parental and (**B**) U87 MG/HNR cells. (**C**) High magnification (20×) of a U87 MG/HNR tumor illustrating frequent mitotic figures. (**D**) High magnification (20×) of a U87 MG/HNR tumor showing regions of necrosis. (**E**) High magnification of a U87 MG/HNR tumor (20×) showing an example of a blood pool. (**F**) High magnification (20×) of a U87 MG/HNR tumor with an area of serous accumulation and poorly-organized extracellular matrix deposition.

**Figure 4 cancers-15-00460-f004:**
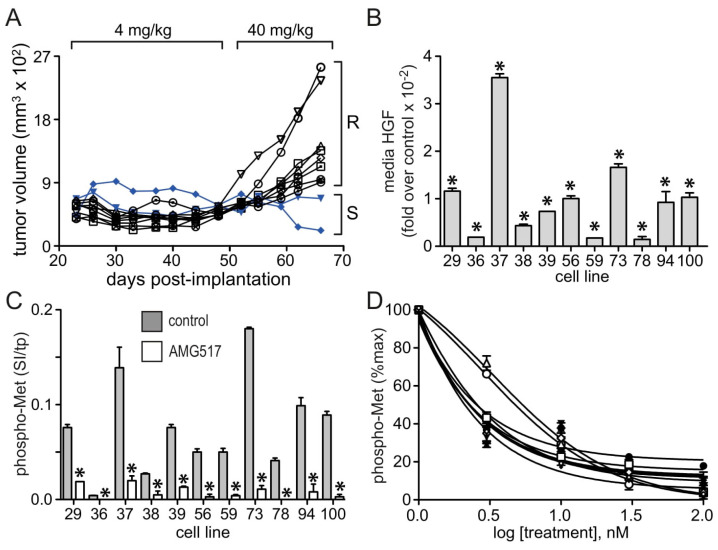
**Acquired resistance to rilotumumab by U87 MG xenografts in vivo.** (**A**) U87 MG tumor xenografts grown in mice (n = 10) treated with rilotumumab at 4 mg/kg until day 48 post-implantation and thereafter with 40 mg/kg displayed drug sensitivity (“S”) or acquired drug resistance (“R”) by day 68. (**B**) Mean HGF content (corrected for total cell protein; ±SD, n = 3), expressed as fold over the parental U87 MG cell line (control), in 24 h media conditioned by 11 cultured cell lines derived from rilotumumab-resistant U87 MG tumor xenografts that were generated as described in panel a. (**C**) Phospho-Met content (mean signal intensity/mg total protein ± SD, n = 3) in 11 cultured cell lines derived from rilotumumab-resistant U87 MG tumor xenografts in the absence (gray bars) or presence (white bars) of 100 nM AMG517. (**D**) Dose-dependent inhibition of phospho-Met (mean % maximum ± SD, n = 3) in 11 cultured cell lines derived from rilotumumab-resistant U87 MG tumor xenografts by the selective small-molecule Met kinase inhibitor AMG517. Asterisks indicate a significant difference from control (*p* < 0.05).

**Figure 5 cancers-15-00460-f005:**
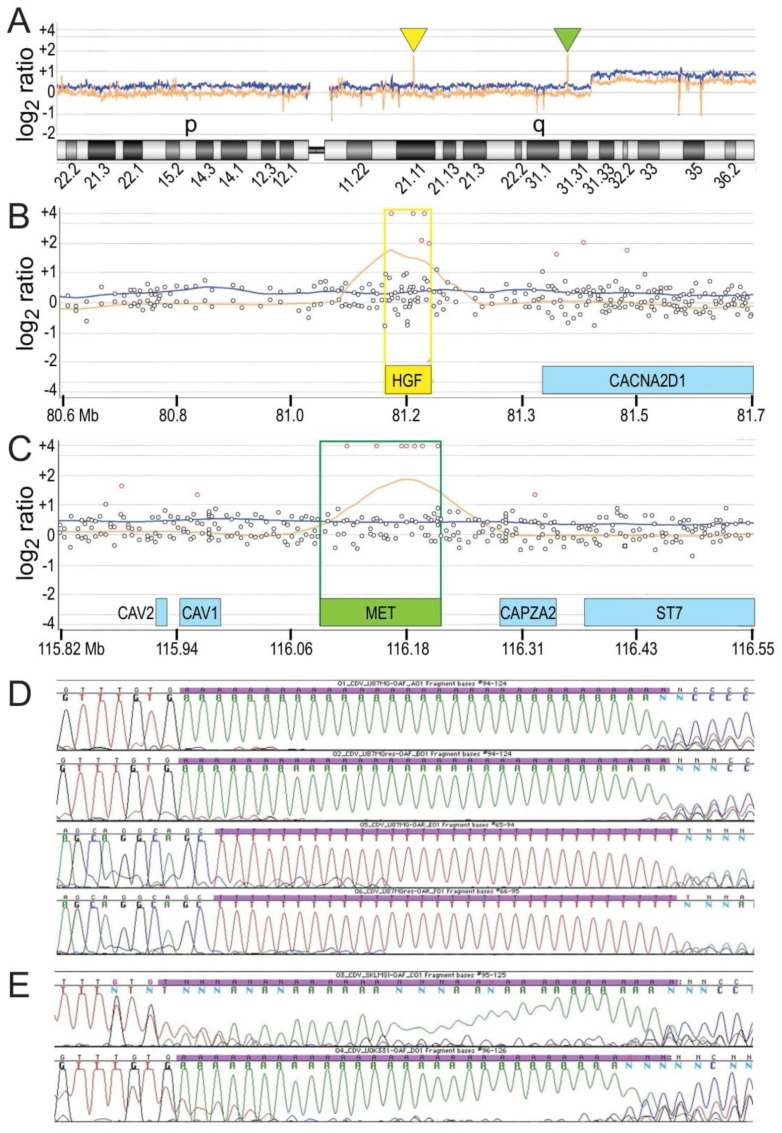
***HGF* and *MET* gene amplification and *HGF* gene promoter DATE region analysis.** (**A**) View of human chromosome 7 CGH microarray analysis results showing moving averages of probe intensity for U87 MG (blue line) or U87 MG/HNR cells (tan line). Inverted triangles indicate the positions of *HGF* (yellow) and *MET* (green) genes. Horizontal lines above and below the center indicate probe intensities corresponding to whole copy number changes. (**B**,**C**) Zoomed view of chromosome 7 regions encoding the genes for *HGF* ((**B**), yellow box) and *MET* ((**C**), green box). Other nearby gene loci are indicated by blue boxes. As in (**A**), moving averages of probe intensities for U87 MG (blue) or U87 MG/HNR cells (tan) are shown. Circles indicate individual probes with intensity values corresponding to gain or loss of less than one gene copy (black) or greater than one gene copy (red). (**D**) DNA sequencing chromatogram encompassing the DATE region in the *HGF* gene promoter. Coding strand sequence (green) for U87 MG DNA (top panel) and U87 MG/HNR DNA (second panel from the top) and non-coding strand sequences for each cell line (third and fourth panels from the top, respectively) shown normal DATE region length in both U87 MG and U87 MG/HNR cells. (**E**) Control samples for DATE sequence truncation (coding strands only) obtained from the leiomyosarcoma cell line SK-LMS-1 (**top**) and for normal DATE region length obtained from the clear cell renal cell carcinoma cell line UOK331 (**bottom**).

**Figure 6 cancers-15-00460-f006:**
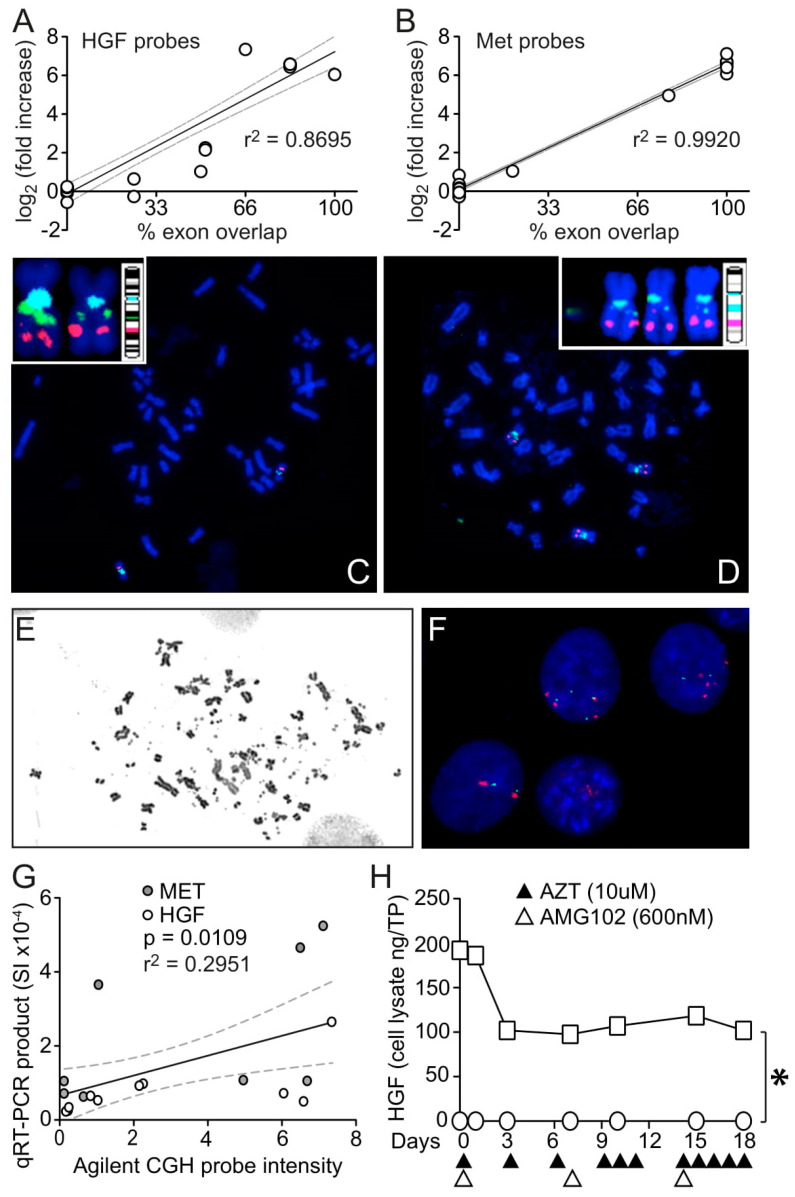
*HGF* and *MET* gene CGH array probe analysis, FISH, qRT-PCR and HGF protein reduction by AZT indicate that amplification involved reverse transcription. Scatter plot of intensity scores (*y*-axis, log_2_ (fold increase)) from individual CGH array probes for *HGF* (**A**) or *MET* (**B**) vs. percent overlap between probe sequence and matching complementary gene exon sequence (*x*-axis). Linear regression analysis lines (black), 95% confidence intervals (dashed gray lines) and r^2^ values are shown. FISH analysis of U87 MG (**C**) and U87 MG/HNR (**D**); *MET* probes are red, *HGF* probes are green and centromeric probes are cyan. Insets in each panel show copies of chromosome 7 adjacent to ideograms of probe locations. Inset for U87 MG/HNR (**D**) shows double-minute chromosomes positive for *HGF* only at left. (**E**) U87 HNR DNA analyzed for metaphases after being cultured for 4 weeks in the presence of rilotumumab; numerous double minute (DM) chromosomes and fragments are visible. (**F**) FISH probes for *HGF* (green) and *MET* (red) hybridized with interphase chromatin show numerous copies of each gene. (**G**) Scatter plot of Agilent CGH array probe intensities for *HGF* and *MET* genes in U87 MG/HNR (*x*-axis) vs. relative content of qRT-PCR products generated using primers corresponding to CGH probe sequences (*y*-axis). Linear regression analysis (black line), 95% confidence limits (gray dashed lines), r^2^ and *p* values of non-linearity are shown. (**H**) HGF protein content (ng/mg total protein, *y*-axis) in samples (n = 3) of cultured U87 MG (circles) or U87 MG/HNR cells (squares) grown for 18 days (*x*-axis) in the presence of added rilotumumab (open triangles below *x*-axis) and AZT (closed triangles below *x*-axis). Error bars (SD) at all time points are not visible because they are smaller than the symbol size. Asterisks indicate a significant difference from control (*p* < 0.05).

**Figure 9 cancers-15-00460-f009:**
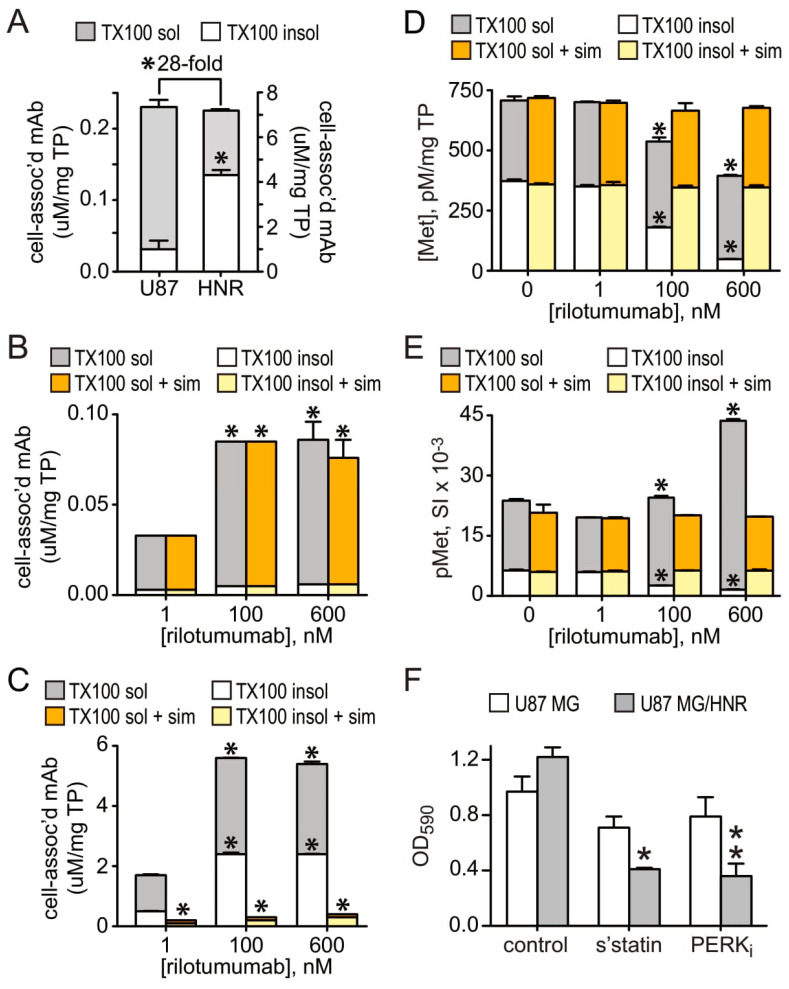
**Qualitative and quantitative changes in vesicular trafficking accompany rilotumumab resistance.** As in Figure 8, cells were extracted with ice-cold buffer containing TX-100 detergent, protease and phosphatase inhibitors before centrifugation to separate soluble (“sup”, left, or “super” right) and insoluble (“pel”, left, or “pellet”, right) cell fractions; insoluble fractions were subsequently solubilized in buffer containing 1% SDS prior immunoassay analysis (see Methods). (**A**) Cell-associated mAb (mean uM rilotumumab/mg total protein ± SD, n = 3) measured in U87 MG (“U87”, left *y*-axis) or U87 MG/HNR (“HNR”, right *y*-axis) cell Triton X-100 extracts separated into soluble (gray bars) and insoluble (white bars) fractions prepared after 24 h rilotumumab exposure. (**B**,**C**): Cell-associated mAb (mean uM rilotumumab/mg total protein ± SD, n = 3) in Triton X-100 soluble (gray or orange bars) and insoluble (white or yellow bars) fractions prepared from U87 MG cells (**B**) or U87 MG/HNR cells (**C**) exposed for 16 h to rilotumumab at the indicated concentrations in the absence (gray/white bars) or presence (yellow/orange bars) of simvastatin (2 mM). (**D**,**E**): Met protein content ((**D**), mean pM/mg total protein ± SD, n = 3) and phosphoMet content ((**E**), mean signal intensity/mg total protein ± SD, n = 3) in U87 MG/HNR cell-derived Triton X-100 extracts separated into soluble (gray or orange bars) and insoluble (white or yellow bars) fractions exposed for 16 h to rilotumumab at the indicated concentrations in the absence (gray/white bars) or presence (yellow/orange bars) of simvastatin (2 mM). (**F**) Soft agar colony formation by U87 MG (clear bars) or U87 MG/HNR cells (gray bars) left untreated (“control”) or treated with simvastatin (“s’statin”) or the selective PERK inhibitor GSK2656157 (“PERK_i_”). Values are mean ± SD from triplicate samples, * (*p* = 0.0024) and ** (*p* = 0.001) indicate statistical significance as determined by unpaired *t*-test with Welch’s correction. Asterisks indicate a significant difference from control (*p* < 0.05).

## Data Availability

Data presented in this study that are not contained in the article are available on request from the corresponding author. The data are not publicly available due to agreements set forth in NIH CRADA No. 01943 between the National Cancer Institute and Amgen, Inc.

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
