# Peer review of "Rilotumumab Resistance Acquired by Intracrine Hepatocyte Growth Factor Signaling"

_cancers, 2023, doi:10.3390/cancers15020460_

Round 1
Reviewer 1 Report (Previous Reviewer 2)
The Authors have done a great job in this manuscript.
Author Response
On behalf of the authors I thank the reviewers for the thorough critiques of our manuscript. We are pleased that our revisions have met with your expectations and believe they have improved our work.
Reviewer 2 Report (Previous Reviewer 1)
The authors answered to all my comments. Asterisks in all figures were still missing in the new version of the manuscript.
Author Response
We thank the Reviewer for their thorough critique of our work and we are happy that our revisions have met expectations. We are certain the revisions have improved our work. We will confirm that the asterisks denoting significant differences are present on the final figures.
This manuscript is a resubmission of an earlier submission. The following is a list of the peer review reports and author responses from that submission.
Round 1
Reviewer 1 Report
In this manuscript, the authors developed a new resistance U87 cell lines to rilotumumab to understand and eventually anticipate acquired resistance to this drug in glioblastoma. This new cell line has a persistent MET pathway activated and are more sensitive to MET kinase inhibitor and inhibition of ER stress. The study is well constructed, and results are very interesting. I have a few comments and questions which may be useful to complete this study.
Materials and Methods
· Generally, to develop resistant cell lines, concentration of the drug is increased over time. Can the authors explain why they decided to use only one concentration?
· Why the authors decided to stop resistant cell line development to 90 days?
· The incubation time is different in M&M (90 days) and in the first paragraph of results (120 days), which one is the correct one?
Figures
· In all figures the resolution is not higher enough. Please modify the figures to have a better resolution.
· I think it will be easier to follow histograms/graph by adding appropriate legends instead of describing only in the legends (Fig. 1E – H; 2B; 3B – C; 4A – D, 7D).
· Significant asterisks are missing in all figures.
· Please add scalebar in all pictures
· AMG517 is not described in the reference 31. Why did the authors decided to use this inhibitor?
· cDNA sequences derived from HGF and MET mRNA transcripts were 356 normal in both the parental and resistant cell lines (data not shown). Should be in Fig Sup.
· Figure 1:
i. It is essential to measure IC50 of these two cell lines to confirm resistance. Please add a clonogenic assay to confirm this.
ii. Please describe the morphology difference between U87 MG and HNR
iii. Harmonization needs to be done in the western blot in C. (same order between condition). What represents the CTL? It will be more impactful to add flow cytometry to measure cell death/add PARP cleaved measurement.
iv. The authors might replace this data with a doubling time measurement in D.
v. Did the authors measure if the resistance is reversible?
· Figure 2:
i. It was complicated to follow this figure. I suggest modifying the order these graphs: 2A should be 2C, 2B should be 2A and 2C should be 2B.
ii. Can the authors explain why there no difference in tumor volume between 0.5, 1.0 and 3.0e6 cells?
iii. Legends order in 2D in the same as 2A, please harmonize.
· Figure 4:
i. It was a very good idea to also developed resistant cell lines in mice, I really appreciate this figure.
ii. Did the authors also select more sensitive cell lines?
iii. Please modify compound A in the legend by AMG517
· Section 3.5:
i. It might be interesting to prepare a complete main or supplemental figure for this section to illustrate the results
Author Response
REVIEWER #1 COMMENTS TO AUTHORS:
In this manuscript, the authors developed a new resistance U87 cell lines to rilotumumab to understand and eventually anticipate acquired resistance to this drug in glioblastoma. This new cell line has a persistent MET pathway activated and are more sensitive to MET kinase inhibitor and inhibition of ER stress. The study is well constructed, and results are very interesting. I have a few comments and questions which may be useful to complete this study.
Materials and Methods
- Generally, to develop resistant cell lines, concentration of the drug is increased over time. Can the authors explain why they decided to use only one concentration?
AUTHOR REPLY:
We thank the reviewer for bringing this matter to our attention and apologize: a lower dose was used initially then increased for the remainder of the treatment period. This information was mistakenly omitted as part of an effort to meet the word limits of another journal. We have revised the Material and Methods text (section 2.1, lines 164-167) and the Results text (section 3.1, lines 344-347) of the revised version of the manuscript to state that a lower dose (100 nM) of rilotumumab was used for the first 5 days and it was then increased to 600 nM for a total treatment period of 120 days to generate the resistant cell lines. A lower initial drug concentration was also used to generate resistant tumor xenografts in mice as shown in Figure 4A.
- Why the authors decided to stop resistant cell line development to 90 days?
AUTHOR REPLY:
The cells acquired complete rilotumumab resistance within 120 days. Observable cell death and re-growth occurred within the first 90 days. Between 90 and 120 days the cultures had a stable growth rate, so waiting longer than 120 days to characterize the treated cells was unnecessary.
- The incubation time is different in M&M (90 days) and in the first paragraph of results (120 days), which one is the correct one?
AUTHOR REPLY:
We apologize for the confusion, the total period of exposure to rilotumumab was 120 days. The mention of 90 days in Materials and Methods has been corrected in the revised manuscript (section 2.1, lines 164-167).
Figures
- In all figures the resolution is not higher enough. Please modify the figures to have a better resolution.
AUTHOR REPLY:
We apologize: we reduced the resolution of the original images to make a single composite figure file that complied with the file size limit for uploading to the Journal's website. Our motive to provide a single composite file was to guarantee that the figures would be ordered correctly in the single PDF file prepared for reviewers. For the revised manuscript we will upload all of the figure files individually to provide higher resolution.
- I think it will be easier to follow histograms/graph by adding appropriate legends instead of describing only in the legends (Fig. 1E – H; 2B; 3B – C; 4A – D, 7D).
AUTHOR REPLY:
Figures 1E through 1G use a different shade of gray for each cell line (the same in all panels); because the cell lines are identified immediately below the x-axis, we believe that adding a graphic legend for these panels is unnecessary. In Figure 1H the cell lines are not identified, so we have added a graphical legend as suggested. Figure 2B did contain a graphical legend; perhaps this did not upload properly, but we will confirm that it is present in the revised manuscript. Figure 3 contains only micrographs; we will add scale bars to these photos as suggested below. In Figure 4, each line (4A) or bar (4B) represents a unique xenograft tumor (4A) or tumor derived cell line (4B); because these are not referred to individually in the text, we believe that graphical legends for these panels are unnecessary. We have added a graphical legend to Figure 4C as suggested, since the bar shading is only otherwise defined in the legend. Figure 4D is similar to 4A in that the curves are not referred to individually in the text, so we do not identify them graphically.
- Significant asterisks are missing in all figures.
AUTHOR REPLY:
We have added these to all figures where appropriate (Figures 1, 2, 4, 6, 7, 8 and 9).
- Please add scalebar in all pictures
AUTHOR REPLY:
Scale bars have been added to all panels in Figure 3. Scale bars are not typically used in human karyotypes (Figure 6C – F).
- AMG517 is not described in the reference 31. Why did the authors decided to use this inhibitor?
AUTHOR REPLY:
We apologize for the citation error. The reference has been corrected to Boezio et al., where AMG517 is identified as compound #22. The compound #22 identifier is included with the in-text citations in the Material and Methods (lines 159-160) and Results (lines 364-365) of the revised manuscript.
This inhibitor was used because it had several desirable features as a small molecule MET kinase inhibitor. As described in Boezio et al. [31], it has IC50 values of 5 nM in biochemical assays and 7 nM in cellular assays, thus it is very potent relative to other compounds in its series and the similarity of these IC50 values indicated good access to the intracellular compartment. Many other compounds in this series also exhibited significant time-dependent inhibition (TDI) of CYP3A4 (measured as CYP3A4 IC50 with vs. without a 30 min pre-incubation with liver microsomes), whereas AMG517 did not. Since CYP3A4 metabolizes ~50% of drugs used in humans, TDI raises concerns for potential drug-drug-interactions in combination cancer therapies and is therefore considered undesirable in clinical candidates.
- cDNA sequences derived from HGF and MET mRNA transcripts were normal in both the parental and resistant cell lines (data not shown). Should be in Fig Sup.
AUTHOR REPLY:
Because these sequences were identical to UniProt P08581-1 (for MET) and UniProt P14210-1 (for HGF) we have stated these complete identities in the Results text (lines 506-507) rather than duplicate the known sequences in the Supplement.
- Figure 1:
- It is essential to measure IC50 of these two cell lines to confirm resistance. Please add a clonogenic assay to confirm this.
AUTHOR REPLY:
The IC50 value for inhibition of U87 MG growth by rilotumumab was published (32 nM, Table 1 in cited reference 22) by co-authors of this manuscript using the same banked cell stock as used for the work described here. A comparable IC50 value for U87 MG HNR cannot be computed because there is no inhibition of cell growth by rilotumumab, i.e. the proliferation rate shown in Figure 1D is not significantly different in the presence or absence of 600 nM rilotumumab. Similarly, an IC50 for inhibition of MET kinase activation by rilotumumab cannot be determined in U87 MG HNR because treatment actually increases phosphoMET (pMET) in a dose dependent manner (Figure 9E).
- Please describe the morphology difference between U87 MG and HNR
AUTHOR REPLY:
The original Results text (lines 347-351) contained the following description:
“Resistant cells (designated U87 MG/HNR for HGF Neutralization Resistant) displayed altered morphology (Fig. 1A, B) and were noticeably less adherent to plastic or extra-cellular matrix substrata in culture than the parental cell line. A relatively high number of floating cells also suggested an increased rate of cell death over the parental cell line.”
The text has been amended as follows (lines 347-353 in the revised manuscript):
“Resistant cells (designated U87 MG/HNR for HGF Neutralization Resistant) displayed altered morphology in 2D culture relative to the parental cell line (Fig. 1A, B). Resistant cells were modestly but consistently smaller, had fewer and shorter extended processes, and had fewer cell-cell interactions (visualized as cell clumping) than parental cells. Resistant cells were also noticeably less adherent to plastic or extra-cellular matrix substrata than the parental cell line. A relatively high number of floating cells also suggested an increased rate of cell death over the parental cell line.”
iii. Harmonization needs to be done in the western blot in C. (same order between condition). What represents the CTL? It will be more impactful to add flow cytometry to measure cell death/add PARP cleaved measurement.
AUTHOR REPLY:
We believe that having the same order of sample loading in Figure 1C will not add information or otherwise improve the figure. The control lanes contain positive control samples for cleaved or intact caspase 3 supplied by the antibody manufacturer. We have revised the Legend for Figure 1C accordingly (lines 384-385).
Our manuscript contains 9 figures with a total of 50 panels as well as 3 tables. Although there are more ways to characterize basic features of resistant cell line than are shown in Figures 1-4, we prioritized further work to define the resistance mechanism over additional ways to measure cell death.
- The authors might replace this data with a doubling time measurement in D.
AUTHOR REPLY:
Doubling time is a single number, and is therefore limited in showing differences in exponential growth rates. Such a change would save relatively little space in our manuscript, so we prefer to keep Figure 1D as shown.
- Did the authors measure if the resistance is reversible?
AUTHOR REPLY:
The resistant cell line was routinely maintained in rilotumumab. No systematic quantitative analysis of resistance reversibility has been undertaken.
- Figure 2:
- It was complicated to follow this figure. I suggest modifying the order these graphs: 2A should be 2C, 2B should be 2A and 2C should be 2B.
AUTHOR REPLY:
We thank the reviewer for this suggestion and agree that the recommended panel order is easier to follow. The panel order has been changed as recommended in the revised Figure 2 file (R1), and appropriate revisions have been made to the Results text (lines 400-410) and the Figure 2 legend (lines 416-450).
- Can the authors explain why there no difference in tumor volume between 0.5, 1.0 and 3.0e6 cells?
AUTHOR REPLY:
There are differences in the mean tumor volumes, but they are not significant at 10 or 13 days. The differences increase with time, but having established the significant difference between the resistant and parental cell lines at day 13, there was no need to keep animals on the study as required by animal use ethics guidelines. At day 13 the mice that received 3 million cells per site had to be euthanized per the same guidelines, illustrating another difficulty with an extended study. To underscore the remarkable magnitude of U87 MG HNR xenograft growth rate, I will note that the senior co-authors of this manuscript (Drs. Coxon, Burgess, and Bottaro), each with over 3 decades of animal research experience, have never observed xenografts from any tumor cell line that grew as fast as U87 MG HNR.
iii. Legends order in 2D in the same as 2A, please harmonize.
AUTHOR REPLY:
This has been revised as suggested.
- Figure 4:
- It was a very good idea to also developed resistant cell lines in mice, I really appreciate this figure.
AUTHOR REPLY:
We thank the reviewer for this comment; it was important to us as well.
- Did the authors also select more sensitive cell lines?
AUTHOR REPLY:
This was not done.
iii. Please modify compound A in the legend by AMG517
AUTHOR REPLY:
This had been corrected.
- Section 3.5:
- It might be interesting to prepare a complete main or supplemental figure for this section to illustrate the results
AUTHOR REPLY:
We considered this suggestion carefully and concluded that a comprehensive schematic would be very time-consuming to construct and its complexity would be quite susceptible to misinterpretation. Ultimately we felt that the effort required would be better used for experiments to follow-up the project’s most intriguing new findings.

Reviewer 2 Report
The study suggests that rilotumumab resistance was acquired through a distinctive mechanism that associated significant HGF overproduction and misfolding, endoplasmic reticulum (ER) stress-response signaling, and redirected vesicular trafficking. This mechanism successfully sequestered rilotumumab from native HGF and activated Met while also producing misfolded HGF. MET and HGF gene augmentation was also noted, along with the rapid acquisition of intron-free, reverse-transcribed copies in DNA. These adjustments enhanced cell survival under stressful circumstances and allowed the Met pathway to remain activated. Overall this is a well-designed study that included both in vivo and in vitro models, however, it needs some corrections to substantiate the outcomes.
1-In addition to the pharmacological inhibitor or MET, The authors also need to confirm the role of MET in using Si-RNA or sh-RNA in vitro models.
2-Since misfolded HGF protein may lead to the inhibition of HGF ubiquitination in tumor cells can alter the expression level of EGFR and regulate cell growth and survival may and cause resistance to Rilotumumab. The author needs to also confirm that misfolded HGF leads to increased cellular senescence or Autophagy and their role in cancer cell survival and Rilotumumab resistance.
3-Did the authors try Rilotumumab in combination with Senolytic drugs or drugs inducing polyubiquitination in these cancer cells?
Author Response
REVIEWER #2 COMMENTS TO AUTHORS:
The study suggests that rilotumumab resistance was acquired through a distinctive mechanism that associated significant HGF overproduction and misfolding, endoplasmic reticulum (ER) stress-response signaling, and redirected vesicular trafficking. This mechanism successfully sequestered rilotumumab from native HGF and activated Met while also producing misfolded HGF. MET and HGF gene augmentation was also noted, along with the rapid acquisition of intron-free, reverse-transcribed copies in DNA. These adjustments enhanced cell survival under stressful circumstances and allowed the Met pathway to remain activated. Overall this is a well-designed study that included both in vivo and in vitro models, however, it needs some corrections to substantiate the outcomes.
1-In addition to the pharmacological inhibitor or MET, The authors also need to confirm the role of MET in using Si-RNA or sh-RNA in vitro models.
AUTHOR REPLY:
We do not see the need to further confirm the role of MET: we have no doubt that MET is required for the HGF dependence of U87 MG for survival, proliferation and tumorigenicity (cited references 22, 25-28 and others). In the entire literature of HGF/MET signaling (>10,000 original papers since 1991, when the ligand-receptor relationship was established), we are unaware of any evidence that HGF can stimulate cell proliferation, motility or morphogenesis in the absence of MET. Rilotumumab resistant cells remain dependent on MET signaling for MET kinase activation (Figures 1, 4), tumorigenicity (Figure 2) and growth in culture (not shown).
MET tyrosine kinase inhibitors have been in clinical development for over 20 years, which has yielded selective, high affinity agents such as AMG517 and considerable knowledge of their effects in model systems and animals, including humans. These two important features enable pharmacological MET kinase inhibition to be used more precisely – in model systems and animals - than most, if not all, methods to eliminate MET expression. And as importantly, eliminating MET transcripts and/or protein is biologically different than kinase inhibition, and likely to cause more extensive intracellularly disruption that is predictably more difficult define. Our well-informed use of AMG517 exerted the expected effects in all of our studies: signaling blockade, inhibition of growth and inhibition of tumorigenesis, effects that are entirely consistent with an essential role of MET in HGF signaling.
2-Since misfolded HGF protein may lead to the inhibition of HGF ubiquitination in tumor cells can alter the expression level of EGFR and regulate cell growth and survival may and cause resistance to Rilotumumab. The author needs to also confirm that misfolded HGF leads to increased cellular senescence or Autophagy and their role in cancer cell survival and Rilotumumab resistance.
AUTHOR REPLY:
We have no indication that EGFR signaling plays a role in the acquired resistance to rilotumumab by U87 MG HNR. EGFR signaling is not required for proliferation or tumorigenicity by the parental U87 MG cells (cited reference 25 and others), but these cells are completely dependent on HGF/Met autocrine signaling for these activities (cited references 22, 25 – 28 and others). EGFR signaling is not sufficient for tumorigenicity by U87 MG HNR, since AMG517 can completely block tumor xenograft growth (Figure 2) and this compound does not block EGFR signaling (Peterson et al., 2015, PMID 25699405, Table S2). Our microarray analysis indicates that EGFR expression is reduced 1.88-fold in the resistant cell line vs. parental (Supplement Table 1), also consistent with not enabling rilotumumab resistance.
We did not assert that HGF misfolding leads to increased programmed cell death in U87 MG HNR. We hypothesized that the resistant line displayed a higher rate of cell death based on observation by light microscopy (Results lines 351 – 353) and showed increased caspase 3 cleavage (Figure 1C) as further evidence of increased cell death for U87 MG HNR vs the parental U87 MG. We also note that this increased cell death does not compensate in any way for the proliferation rate acquired with resistance, which is many fold higher than the parental cell line (Figure 1D). We do link HGF misfolding to increased Unfolded Protein Response (UPR) signaling (Figure 8A and B), intracellular retention of HGF protein (Figure 8C and D), and acquired sensitivity of anchorage-independent growth to PERK inhibition with rilotumumab resistance (Figure 9F). Further defining the mechanism of increased cell death would not contradict the evidence of ER stress and UPR signaling, and while potentially interesting, it was not a priority.
3-Did the authors try Rilotumumab in combination with Senolytic drugs or drugs inducing polyubiquitination in these cancer cells?
AUTHOR REPLY:
We have not done experiments with Senolytic drugs or drugs inducing polyubiquitination in the resistant or parental cell lines.
